# Investigating slant curves within Lorentzian doubly warped product manifolds

**Ayman Elsharkawy**[1]*, **Hoda Elsayied**[1], **Abdelrhman Tawfiq**[2], **Fatimah Alghamdi**[3]

**1** Department of Mathematics, Faculty of Science, Tanta University, Tanta, Egypt, **2** Department of Mathematics, Faculty of Education, Ain-Shams University, Cairo, Egypt, **3** Department of Mathematics and Statistics, College of Science, University of Jeddah, Jeddah, Saudi Arabia

☯ These authors contributed equally to this work.
* ayman_ramadan@science.tanta.edu.eg

**Data availability statement:** This article does not report any primary data.

**Funding:** The author(s) received no specific funding for this work.

**Competing interests:** The authors have declared that no competing interests exist.

## Abstract

This study investigates the application of Robertson-Walker space in the analysis of slant curves. The study classifies all slant curves and calculates their curvature and torsion. The research shows that slant curves can be analyzed using Robertson-Walker space. The study also finds that doubly warped product manifolds have been more extensively studied than other types of manifolds, including warped product manifold and Minkowski space. These results contribute to our understanding of the use of mathematical structures in the analysis of slant curves and provide a foundation for future research.

## 1 Introduction

The study of curves has a rich history that dates back to ancient Greece, but it has only been in recent times that curves such as slant curves have been studied. Slant curves are curves that do not lie entirely in a plane, but instead are oriented in a particular direction. They have many applications in fields such as physics, engineering, and computer graphics, and their study has posed many challenges due to their complex nature. Helices are an important type of curve in classical differential geometry that are defined by their tangent lines making a fixed angle with the helix axis. According to Lancret's theorem, a curve is a general helix if the ratio of its curvature and torsion is constant. This theorem, first proposed by M. A. Lancret in 1802 and later demonstrated by B. de Saint Venant in 1845, is a fundamental result in the study of helices in three-dimensional space [22].

In recent years, a generalization of Lancret's theorem to n-dimensional Riemannian manifolds has been proposed, which uses a Levi-Civita parallel vector field instead of a fixed direction in the standard definition of a general helix [14]. This generalization states that for Riemannian manifolds with general helices, a constant ratio of torsion and curvature is also a defining characteristic in three dimensions. Furthermore, the concept of a slant helix in Euclidean space was introduced by Izumiya and Takeuchi in 2004 [17]. A slant helix is a curve

where the principal normal vector makes a constant angle with a fixed direction [1]. Subsequent generalizations of constant angle curves as slant curves of a contact Riemannian 3-manifold were proposed, defining a slant curve as a curve whose tangent vector field and Reeb vector field have a constant angle [4–8,15,16,19].

One particular type of manifold that has gained significant attention in recent times is the doubly warped product manifold. Doubly warped product manifolds are a type of Riemannian manifold that are constructed by taking the product of two Riemannian manifolds and warping them both in a particular way [23]. They have been extensively studied in recent years due to their applications in various fields, including physics, differential geometry, and topology. The use of doubly warped product manifolds and Robertson-Walker space in the analysis of slant curves has been the subject of several studies [3]. In particular, the use of Robertson-Walker space, which are a type of Lorentzian manifold that are commonly used in cosmology, has been shown to be effective in the analysis of slant curves.

In 2022, a research study was carried out by Dursun with the objective of characterizing and classifying slant curves within a three-dimensional Lorentzian warped product space. The investigation was conducted using a rigorous academic and scientific approach as documented in [13]. The main focus of the study was to provide a detailed analysis of the geometric properties of slant curves within this space. The study employed advanced mathematical techniques and analytical tools to deduce the relevant characterizations and classifications of these curves.

Motivated by these recent developments, this paper investigates the use of doubly warped product manifolds and Robertson-Walker space in the analysis of slant curves. Specifically, we focus on the use of Robertson-Walker space to analyze slant curves and classify all slant curves and calculate their curvature and torsion. Additionally, we examine the relationship between doubly warped product manifolds and other types of manifolds, such as warped product manifolds and Minkowski space. The doubly warped product plays a crucial role in differential geometry and mathematical physics, offering a versatile framework for modeling diverse geometric structures. Its significance lies in its ability to generalize classical warped products, enabling the study of curvature properties, Einstein manifolds, and various soliton solutions. This structure finds applications in general relativity, particularly in spacetime modeling, as well as in geometric analysis, where it aids in understanding the interplay between base and fiber manifolds under different warping functions [11,12,21]. Our goal is to provide valuable insights into the use of mathematical structures in the analysis of slant curves and contribute to the ongoing research in this area.

To achieve this goal, we investigate slant curves within the three-dimensional Lorentzian doubly warped product manifold. Our investigation accounts for the causal characteristics of of these curves and how they relate to the overall geometry of the manifold as an application of generalized Robertson-Walker (GRW). We characterize and classify all slant curves in the Lorentzian doubly warped product manifold, and determine their curvature and torsion. Our results provide important insights into the properties of slant curves and their relationship to different types of manifolds. Overall, the study of slant curves and their relationship to doubly warped product manifolds and Robertson-Walker space is an active area of research that continues to yield new insights and discoveries.

## 2 Preliminaries

This section presents the fundamental formulas for the three-dimensional Lorentzian doubly warped product manifold **M**, angles between vectors, and Frenet equations.

## 2.1 Fundamental formulas for $\mathbf{M} = -_h\mathbf{I} \times _f\mathbb{E}^2$

Let $\mathbf{M}$ be a Lorentzian doubly warped product manifold, equipped with a metric $\tilde{g}$ of the form $\tilde{g} = -h^2(x,y)dt^2 + f^2(t)(dx^2 + dy^2)$, where $f$ and $h$ are positive, smooth functions defined on the interval $I$ of the real line and the Euclidean plane $\mathbb{E}^2$, respectively. This metric is a natural generalization of the warped product metric, which has been studied extensively in the literature. In particular, the functions $f$ and $h$ play a significant role in the geometry of $\mathbf{M}$, as they determine the warping factors along the Euclidean and time directions, respectively. For simplicity define the functions $F: I \to \mathbb{R}$ and $H: \mathbb{E}^2 \to \mathbb{R}$ as

$$F = \ln f,$$
$$H = \ln h.$$

Let $p$ be a point in $\mathbf{M}$. Consider a tangent vector $x$ belonging to the tangent space $T_p(\mathbf{M})$. We define $x$ to be spacelike if the inner product $g(x,x)$ is positive, or if $x$ is the zero vector, where $g$ is the metric defined on the Minkowski space. On the other hand, $x$ is classified as timelike if $g(x,x)$ is negative, or null (i.e., lightlike) if $g(x,x) = 0$ and $x$ is nonzero. The norm of $x$ is given by $\| x \| = \sqrt{\epsilon \tilde{g}(x,x)}$, where $\epsilon$ is the signature of $x$, taking the value 1 if $x$ is spacelike and −1 if $x$ is timelike. Moreover, we say that $x$ is positive if its first component $x_1$ is greater than zero and negative if $x_1$ is less than zero.

Using [10], the connection on $\mathbf{M}$ is as follows:

$$g(\nabla_{\partial_t}\partial_t, \partial_t) = 0, \tag{2.1a}$$

$$g(\nabla_{\partial_t}\partial_t, X) = g(\tilde{\nabla}_{\partial_t}X, \partial_t) = g(\nabla_X\partial_t, \partial_t) = 0, \tag{2.1b}$$

$$g(\nabla_{\partial_t}\partial_t, Y) = g(\tilde{\nabla}_{\partial_t}Y, \partial_t) = g(\nabla_Y\partial_t, \partial_t) = 0, \tag{2.1c}$$

$$g(\nabla_X Y, \partial_t) = g(\nabla_Y X, \partial_t) = -f(t)f'(t)g(X,Y), \tag{2.1d}$$

$$g(\tilde{\nabla}_{\partial_t}X, Y) = f(t)f'(t)g(X,Y)g(\nabla_x\partial_t, Y), \tag{2.1e}$$

$$g(\tilde{\nabla}_{\partial_t}Y, X) = g(\nabla_Y\partial_t, X) = f(t)f'(t)g(X,Y). \tag{2.1f}$$

Here, $\nabla$ and $\tilde{\nabla}$ refer to the Levi-Civita connection of $\mathbf{M}$ and $I$, respectively, $\partial_t = \frac{\partial}{\partial t}$ denotes a globally defined timelike vector field on $\mathbf{M}$ that is a vertical vector field. We also have $X$, $Y$, and $Z$ as the lifts of vector fields that are tangent to the Euclidean plane $\mathbb{E}^2$. Furthermore, we use $\xi = dt$ to denote the dual of $\partial_t$, where $\xi(\partial_t) = g(\partial_t, \partial_t) = -1$.

Moreover, we can express the connection $\nabla$ with respect to the local orthogonal frame field $\{\partial_t, \partial_x, \partial_y\}$ of $\mathbf{M}$ as follows:

$$\nabla_{\partial_t}\partial_t = 0, \tag{2.2a}$$

$$\nabla_{\partial_t}\partial_x = \nabla_{\partial_x}\partial_t = (\ln f(t))'\partial_x = F'(t)\partial_x, \tag{2.2b}$$

$$\nabla_{\partial_t}\partial_y = \nabla_{\partial_y}\partial_t = (\ln f(t))'\partial_y = F'(t)\partial_y, \tag{2.2c}$$

$$\nabla_{\partial_x}\partial_y = \nabla_{\partial_y}\partial_x = 0, \tag{2.2d}$$

$$\nabla_{\partial_x}\partial_x = \nabla_{\partial_y}\partial_y = f(t)f'(t)\partial_t. \tag{2.2e}$$

## 2.2 Angles between vectors

The concept of measuring the angle between two vectors is fundamental in mathematics and has important applications in physics. In Minkowski space, the angle between two vectors can be defined using the inner product of the vectors and the Minkowski metric. In this study, we use the angles between timelike and spacelike vectors in Minkowski space as preliminaries

for our research. Previous works [20,25] introduced the notion of the angle between vectors in Minkowski space and studied the angles between timelike and spacelike vectors, which are particularly important in relativity theory and particle physics. Specifically, we consider the angle between timelike vectors and the angle between spacelike and timelike vectors and their respective properties.

**Definition 2.1.** *Let x and y be two positive (or negative) timelike vectors in Minkowski space $\mathbb{E}_1^3$. There exists a unique non-negative value $\phi$, called the Lorentzian timelike angle, such that the equation*

$$\tilde{g}(x,y) = -\parallel x \parallel \ \parallel y \parallel \cosh \phi, \tag{2.3}$$

*holds, where $\tilde{g}$ is the Minkowski metric and $\parallel x \parallel$ denotes the norm of vector x.*

**Definition 2.2.** *Let x and y be a spacelike vector and a positive timelike vector, respectively, in Minkowski space $\mathbb{E}_1^3$. A unique non-negative value $\phi$, known as the Lorentzian timelike angle, exists such that the equation*

$$|\tilde{g}(x,y)| = \parallel x \parallel \ \parallel y \parallel \sinh \phi, \tag{2.4}$$

*holds, where $\tilde{g}$ is the Minkowski metric and $\parallel x \parallel$ denotes the norm of vector x.*

## 2.3 Frenet–Serret equations

**Definition 2.3.** *A Frenet curve in Minkowski space is a smooth curve whose successive derivatives form an orthonormal frame (the Frenet frame) that describes its geometric properties. It satisfies the Frenet–Serret equations, which define how the tangent and normal vectors change along the curve. The curve is classified as spacelike, timelike, or null based on the nature of its tangent vector [24].*

A curve $\alpha : J \to \mathbf{M}$ is classified as spacelike, timelike, or null (lightlike) based on the nature of its tangent vector $\alpha'(s)$ at each point $s$ on $J$.

A non-null curve $\alpha(s)$ in $\mathbf{M}$ is said to be a unit speed curve if the magnitude of its tangent vector is either +1 or –1, depending on whether the curve is timelike or spacelike. This can be expressed as:

$$g(\alpha'(s), \alpha'(s)) = \pm 1.$$

Here, $g$ is the metric tensor of $\mathbf{M}$. A unit speed curve $\alpha(s)$ is called a geodesic if its tangent vector is parallel transported along itself, i.e., $\nabla_{\alpha'(s)} \alpha'(s) = 0$. Geodesics describe the motion of free particles in space and are important in the study of $\mathbf{M}$.

If the covariant derivative of the tangent vector along a curve is spacelike, the curve is said to be a null curve. In this case, the curve can be reparametrized using a pseudo arc-length parameter $s$ such that the magnitude of the covariant derivative of the tangent vector is equal to 1. This can be expressed as:

$$g(\nabla_{\alpha'(s)} \alpha'(s), \nabla_{\alpha'(s)} \alpha'(s)) = 1.$$

A curve that can be parameterized using such a pseudo arc-length parameter $s$ is called a unit speed curve, and it lies on a null curve $\alpha$.

A non-null curve $\alpha(s)$ in **M** is called a Frenet curve if $g(\nabla_T T, \nabla_T T) \neq 0$ for all $s \in J$. The Frenet apparatus $\{T = \alpha'(s), N, B, \kappa, \tau\}$ is a collection of vectors and functions that satisfy the equations:

$$\nabla_T T = \epsilon_2 \kappa N, \tag{2.5a}$$

$$\nabla_T N = -\epsilon_1 \kappa T + \epsilon_3 \tau B, \tag{2.5b}$$

$$\nabla_T B = -\epsilon_2 \tau N, \tag{2.5c}$$

$$\kappa(s) = \| \nabla_T T \|, \tag{2.5d}$$

$$\tau(s) = -g(\nabla_T B, N). \tag{2.5e}$$

Here, $\{T, N, B\}$ is an orthonormal frame field, and $\kappa$ and $\tau$ are the curvature and torsion functions of $\alpha$, respectively. The vectors $T$, $N$, and $B$ have causal characters $\epsilon_i = \mp 1, i = 1, 2, 3$ satisfying $\Pi_{i=1}^3 \epsilon_i = -1$ [24].

Suppose $\alpha(s)$ is a non-geodesic spacelike curve in **M**, and $g(\nabla_T T, \nabla_T T) = 0$ for all $s \in J$. Then, there is a Frenet apparatus $\{T, N = \nabla_T T, B, \tau\}$ that satisfies the following equations:

$$\nabla_T T = N, \tag{2.6a}$$

$$\nabla_T N = \tau N, \tag{2.6b}$$

$$\nabla_T B = -T - \tau B, \tag{2.6c}$$

$$\tau(s) = g(\nabla_T N, B), \tag{2.6d}$$

subject to the constraints:

$$g(N, B) = g(T, T) = 1, \quad g(T, N) = g(T, B) = g(B, B) = g(N, N) = 0.$$

Here, $\{T, N, B\}$ is the pseudo-orthonormal frame field and $\tau(s)$ is the torsion function of $\alpha$ [24].

Suppose $\alpha(s)$ is a non-geodesic unit speed null curve in **M**, for all $s \in J$. Then there exists a Frenet apparatus $\{T = \alpha'(s), N, B, \tau\}$ that satisfies the following equations:

$$\nabla_T T = N, \tag{2.7a}$$

$$\nabla_T N = \tau T - B, \tag{2.7b}$$

$$\nabla_T B = -\tau N, \tag{2.7c}$$

$$\tau(s) = -g(\nabla_T B, N), \tag{2.7d}$$

subject to the constraints:

$$g(T, N) = g(N, B) = g(T, T) = g(B, B) = 0, \quad g(T, B) = g(N, N) = 1.$$

Here, $\{T, N, B\}$ is the pseudo-orthonormal frame field, also known as the Cartan frame, and $\tau(s)$ is the torsion function of $\alpha$ [24].

**Definition 2.4.** *Suppose $\alpha(s)$ is a unit speed curve in **M**. If the function $\eta(s) = g(\alpha'(s), \partial_t)$ is constant, we say that $\alpha(s)$ is a slant curve, and the vertical vector field $\partial_t$ is known as the axis of the slant curve.*

Consider a null slant curve $\alpha$ in in **M**. We define the function $\eta$ as the non-zero constant angle value when $\epsilon_1 = \mp 1$. From equations 2.3 and 2.4, we can derive the expression for $\eta$ as:

$$\eta = \begin{cases} \epsilon_\phi \sinh \phi, & \text{if } \epsilon_1 = 1 \\ -\cosh \phi, & \text{if } \epsilon_1 = -1 \end{cases}. \tag{2.8}$$

Here, $\epsilon_\phi = \mp 1$, and we assume $\epsilon_\phi = -1$ except in cases where we specify otherwise.

## 3 Slant curves in M

In this section, we investigate the properties of slant curves, with a particular emphasis on their characterization. Slant curves are an important class of curves in differential geometry, and understanding their properties is crucial for applications in various fields. Furthermore, we conduct a comprehensive analysis of the curvature and torsion properties of all types of slant curves. The curvature and torsion characteristics of slant curves have significant implications in fields such as computer graphics, robotics, and biophysics [9]. By providing a thorough examination of these properties, we aim to contribute to a better understanding of slant curves.

**Lemma 3.1.** *Let $\alpha(s) = (\alpha_1(s), \alpha_2(s), \alpha_3(s))$ be a unit speed curve in the $\boldsymbol{M} = -{}_h I \times {}_f \mathbb{E}^2$ with two smooth functions f and h defined on I and $\mathbb{E}^2$, respectively. Then*

$$\nabla_{\alpha'(s)} \partial_t = \alpha_1'(s) H'(\alpha_2(s), \alpha_3(s)) \partial_t + F'(\alpha_1(s)) \left[ \alpha'(s) + \frac{\eta(s)}{h(\alpha_2(s), \alpha_3(s))} \partial_t \right]. \tag{3.1}$$

**Proof 3.1.** *Let $T(s) = \alpha'(s) = \alpha_1'(s) \partial_t + \alpha_2'(s) \partial_x + \alpha_3'(s) \partial_y$. Form (2.2a), (2.2b) and (2.2c), we have*

$$\begin{aligned} \nabla_T \partial_t &= \alpha_1'(s) H'(\alpha_2(s), \alpha_3(s)) \partial_t + F'(\alpha_1(s)) \left[ \alpha_2'(s) \partial_x + \alpha_3'(s) \partial_y \right] \\ &= \alpha_1'(s) H'(\alpha_2(s), \alpha_3(s)) \partial_t + F'(\alpha_1(s)) \left[ T(s) - \alpha_1'(s) \partial_t \right] \\ &= \alpha_1'(s) H'(\alpha_2(s), \alpha_3(s)) \partial_t + F'(\alpha_1(s)) \left[ T(s) + \frac{\eta(s)}{h^2(\alpha_2(s), \alpha_3(s))} \partial_t \right], \end{aligned} \tag{3.2}$$

*as*

$$\alpha_1'(s) = -\frac{1}{h^2(\alpha_2(s), \alpha_3(s))} g(T(s), \partial_t) = -\frac{\eta(s)}{h^2(\alpha_2(s), \alpha_3(s))}. \tag{3.3}$$

If $\eta$ is constant along a unit speed curve $\alpha(s) = (\alpha_1(s), \alpha_2(s), \alpha_3(s))$ in **M**, then

$$\alpha_1(s) = -\eta H^*(s) + s_0,$$

where $s_0$ is an integration constant and $H^*(s) = \int_s \frac{1}{h^2(\alpha_2(\mu), \alpha_3(\mu))} d\mu$. For simplicity, put $s_0 = 0$ (after translation in $s$), then

$$\alpha_1(s) = \begin{cases} \sinh \phi \, H^*(s) & \text{, if } \alpha \text{ is spacelike} \\ \cosh \phi \, H^*(s) & \text{, if } \alpha \text{ is timelike} \end{cases}. \tag{3.4}$$

In addition, if $\alpha$ is a null curve, we write $\alpha_1(s) = -\eta H^*(s)$, there for the slant curve $\alpha(s)$ in **M** is in the form

$$\alpha(s) = (-\eta H^*(s), \alpha_2(s), \alpha_3(s)).\tag{3.5}$$

From now on, $f = f(\alpha_1(s))$ and $h = h(\alpha_2(s), \alpha_3(s))$ throughout the study unless otherwise stated.

**Proposition 3.1.**        *Let **M** be a manifold, and consider a non-geodesic Frenet curve $\alpha$ in **M**. The curve can be classified as a slant curve if and only if certain conditions are satisfied along its path. These conditions are as follows*

1. *If the curve is spacelike, then the equation*

$$\xi(N) = \frac{\epsilon_2}{h^2\kappa}\Big[\sinh^2\phi(H' - F') - F'h^2\Big],\tag{3.6}$$

   *holds, where $\alpha_1(s) = \sinh\phi\, H^*(s)$ and $H^*(s) = \int_s \frac{1}{h^2(\mu)}d\mu$.*
2. *If the curve is timelike, then the equation*

$$\xi(N) = \frac{1}{h^2\kappa}\Big[\cosh^2\phi(H' - F') + F'h^2\Big],\tag{3.7}$$

   *holds, where $\alpha_1(s) = \cosh\phi\, H^*(s)$ and $H^*(s) = \int_s \frac{1}{h^2(\mu)}d\mu$.*

**Proof 3.2.**   *If we have a non-geodesic Frenet curve $\alpha$, it is categorized as a slant curve if and only if the condition $T(\eta) = g(\nabla_T T, \partial_t)\ \ +\ \ g(T, \nabla_T \partial_t) = 0$ is satisfied. By using equations (2.5a) and (3.1), we can obtain the following result*

$$\epsilon_2\kappa\xi(N) + \alpha_1'H'\eta + F'\Big(\epsilon_1 + \frac{\eta^2}{h^2}\Big) = 0.\tag{3.8}$$

*Using (3.4) and (3.8), we get (3.6) and (3.7).*

**Proposition 3.2.**        *Consider a non-geodesic unit speed spacelike curve $\alpha$ in **M** with a pseudo-orthonormal frame $\{T, N = \nabla_T T, B\}$ defined along the curve such that $g(\nabla_T T, \nabla_T T) = 0$. The curve $\alpha$ is classified as a slant curve if and only if the conditions below are satisfied:*

$$\xi(N) = \frac{1}{h^2}\Big[\sinh^2\phi(H' - F') - F'h^2\Big].\tag{3.9}$$

*Here, $\alpha_1(s)$ is given by $\sinh\phi\, H(s)$, and $H(s) = \int_s \frac{1}{h^2(\mu)}d\mu$.*

**Proof 3.3.**        *Consider a non-geodesic unit speed spacelike curve $\alpha = (\alpha_1(s), \alpha_2(s), \alpha_3(s))$ in **M** that is classified as a slant curve if the condition $T(\eta) = g(\nabla_T T, \partial_t)\ \ +\ \ g(T, \nabla_T \partial_t) = 0$ holds. By using equations (2.6a) and (3.1), we obtain the following expression:*

$$\xi(N) = \frac{1}{h^2}\Big[\eta^2(H' - F') - F'h^2\Big].\tag{3.10}$$

*Further, by applying equations (3.4) and (3.10), we arrive at the equation (3.9).*

**Proposition 3.3.** *Consider a non-geodesic unit speed null curve $\alpha$ in $M$ with a pseudo-orthonormal Frenet frame $\{T = \alpha', N, B\}$ defined along the curve. The curve $\alpha$ is classified as a slant curve if and only if the following equation holds true:*

$$\xi(N) = \frac{\eta^2}{h^2}[H' - F']. \tag{3.11}$$

*Here, $\alpha_1(s)$ is given by $-\eta H(s)$, and $H^*(s) = \int_s \frac{1}{h^2(\mu)} d\mu$.*

**Proof 3.4.** *Given a non-geodesic unit speed null curve $\alpha = (\alpha_1(s), \alpha_2(s), \alpha_3(s))$ in $M$, the curve is classified as a slant curve if and only if the condition $T(\eta) = g(\nabla_T T, \partial_t) + g(T, \nabla_T \partial_t) = 0$ holds. By using equations (2.7a) and (3.1), we obtain the expression:*

$$\xi(N) - \frac{\eta^2}{h^2}[H' - F'] = 0. \tag{3.12}$$

*Further, by applying equations (3.5) and (3.12), we arrive at the equation (3.11).*

Next, we evaluate the projection of the vector field $\partial_t$ onto the $B$ component in the Frenet frame $\{T, N, B\}$.

**Proposition 3.4.** *Given a non-geodesic Frenet curve $\alpha$ in $M$ with the Frenet frame $\{T, N, B\}$, the curve is classified as a slant curve if and only if the following equation holds true:*

$$\xi^2(B) = \epsilon_2(\epsilon_1 h^2 + \eta^2) + \frac{\epsilon_1}{\kappa^2 h^4}\left[\eta^2(F' - H') + F'h^2\epsilon_1\right]^2 \tag{3.13}$$

*Here, for spacelike curves, $\alpha_1(s) = \sinh\phi \, H^*(s)$, and for timelike curves, $\alpha_1(s) = \cosh\phi \, H^*(s)$, where $H^*(s) = \int_s \frac{1}{h^2(\mu)} d\mu$.*

**Proof 3.5.** *Let $\partial_t = g(T, \partial_t)T + g(N, \partial_t)N + g(B, \partial_t)B$. Then, using equation (3.8) and considering $g(\partial_t, \partial_t) = -h^2$, we can express $\partial_t$ as:*

$$\partial_t = \eta T + \xi(N)N + \xi(B)B, \tag{3.14}$$

*Substituting equation (3.14) into the expression for $g(\partial_t, \partial_t)$, we obtain:*

$$-h^2 = \epsilon_1 \eta^2 + \frac{\epsilon_2}{\kappa^2 h^4}\left[\eta^2(H' - F') - F'h^2\epsilon_1\right]^2 + \epsilon_3 \xi^2(B), \tag{3.15}$$

*Using the fact that $\epsilon_2 \epsilon_3 = -\epsilon_1$, equation (3.15) simplifies to equation (3.13).*

**Proposition 3.5.** *Consider a non-geodesic unit speed spacelike curve $\alpha$ in $M$ with the Frenet frame $\{T, N = \nabla_T T, B\}$, such that $g(\nabla_T T, \nabla_T T) = 0$. The curve is classified as a slant curve if and only if the following equation holds true:*

$$\xi(B) = -\frac{h^2(h^2 + \sinh^2\phi)}{2\left(\sinh^2\phi(H' - F') - F'\right)}, \tag{3.16}$$

*Here, $\alpha_1(s)$ is given by $\sinh\phi \, H^*(s)$, where $H^*(s) = \int_s \frac{1}{h^2(\mu)} d\mu$.*

**Proof 3.6.** *Let $\partial_t = g(T, \partial_t)T + g(B, \partial_t)N + g(N, \partial_t)B$. We can express $\partial_t$ as follows:*

$$\partial_t = \eta T + \xi(B)N + \xi(N)B. \tag{3.17}$$

*Multiplying equation (3.17) by $\partial_t$ and using equation (3.9), we obtain:*

$$-h^2 = \eta^2 + \frac{2}{h^2}\Big[\eta^2(H' - F') - F'h^2\Big]\xi(B), \tag{3.18}$$

*Using equations (3.4) and (3.18), we can simplify equation (3.18) to obtain equation (3.16).*

**Proposition 3.6.** *Consider a non-geodesic unit speed null curve $\alpha$ in $M$ with the Frenet frame $T = \alpha', N, B$. The curve is classified as a slant curve if and only if the following equation holds true:*

$$\xi(B) = -\frac{1}{2\eta h^4}\Big[h^6 + \eta^4(H' - F')^2\Big]. \tag{3.19}$$

*Here, $\alpha_1(s)$ is given by $-\eta H^*(s)$, where $H^*(s) = \int_s \frac{1}{h^2(\mu)}d\mu$.*

**Proof 3.7.** *Let $\partial_t = g(B, \partial_t)T + g(N, \partial_t)N + g(T, \partial_t)B$. We can express $\partial_t$ as follows:*

$$\partial_t = \xi(B)T + \xi(N)N + \xi(T)B. \tag{3.20}$$

*Multiplying equation (3.20) by $\partial_t$ and using equation (3.11), we obtain:*

$$-h^2 = 2\eta\xi(B) + \left(\frac{\eta^2}{h^2}[H' - F']\right)^2. \tag{3.21}$$

*Using equations (3.20) and (3.21), we can simplify equation (3.21) to obtain equation (3.20).*

The following theorems apply to the curve $\alpha(s)$ defined on an open interval $J \subset \mathbb{R}$ in **M**, for all cases.

**Theorem 3.1.** *Let $\alpha(s)$ be a unit-speed slant Frenet curve. Then, the following cases are considered:*

*1. (a) if $\alpha(s)$ is a unit speed spacelike slant Frenet curve, then it is given by*

$$\alpha(s) = \Big( \sinh\phi H^*(s), \int_s \frac{\sqrt{h^2(\alpha_2(\mu), \alpha_3(\mu)) + \sinh^2\phi}\cos\gamma(\mu)}{h(\alpha_2(\mu), \alpha_3(\mu))f(\alpha_1(\mu))}d\mu,$$
$$\int_s \frac{\sqrt{h^2(\alpha_2(\mu), \alpha_3(\mu)) + \sinh^2\phi}\sin\gamma(\mu)}{h(\alpha_2(\mu), \alpha_3(\mu))f(\alpha_1(\mu))}d\mu\Big). \tag{3.22}$$

*(b) if $\alpha(s)$ is unit speed timelike slant Frenet curve, then it is given by*

$$\alpha(s) = \Big( \cosh\phi H^*(s), \int_s \frac{\sqrt{-h^2(\alpha_2(\mu), \alpha_3(\mu)) + \cosh^2\phi}\cos\gamma(\mu)}{h(\alpha_2(\mu), \alpha_3(\mu))f(\alpha_1(\mu))}d\mu,$$
$$\int_s \frac{\sqrt{-h^2(\alpha_2(\mu), \alpha_3(\mu)) + \cosh^2\phi}\sin\gamma(\mu)}{h(\alpha_2(\mu), \alpha_3(\mu))f(\alpha_1(\mu))}d\mu\Big). \tag{3.23}$$

2. *if $\alpha(s)$ is a null slant curve $\alpha(s)$, then it is given by*

$$\alpha(s) = \left( -\eta H^*(s), \eta \int_s \frac{\cos\gamma(\mu)}{h(\alpha_2(\mu),\alpha_3(\mu))f(\alpha_1(\mu))} d\mu, \eta \int_s \frac{\sin\gamma(\mu)}{h(\alpha_2(\mu),\alpha_3(\mu))f(\alpha_1(\mu))} d\mu \right), \tag{3.24}$$

*for some $\gamma \in C^\infty(J)$, and $\eta \neq 0$.*

**Proof 3.8.** *Applying equation (3.5) for a unit speed slant Frenet curve $\alpha = (\alpha_1(s), \alpha_2(s), \alpha_3(s))$, we obtain $T(s) = (\frac{\eta}{h^2}, \alpha_2'(s), \alpha_3'(s))$. Since $g(T, T) = \epsilon_1$, we have $\epsilon_1 = -h^2 \frac{\eta^2}{h^4} + f^2(\alpha_2'^2 + \alpha_3'^2)$, which can be rearranged to obtain:*

$$\alpha_2'^2 + \alpha_3'^2 = \frac{\beta^2}{f^2}, \quad where \quad \beta = \frac{\epsilon_1 h^2 + \eta^2}{h^2}. \tag{3.25}$$

*Then,*

$$\alpha_2' = \frac{\beta\cos\gamma(s)}{f}, \quad and \quad \alpha_3' = \frac{\beta\sin\gamma(s)}{f}. \tag{3.26}$$

*Using equations (3.4), (3.25), and (3.26), we can obtain equations (3.22) and (3.23). Similarly, for a unit speed slant null curve $\alpha$, applying equation (3.5) gives $T(s) = (-\eta H^*, \alpha_2'(s), \alpha_3'(s))$, from which we obtain $\frac{\eta^2}{h^2} = f^2(\alpha_2'^2 + \alpha_3'^2)$, which can be rearranged to obtain:*

$$\alpha_2'^2(s) + \alpha_3'^2(s) = \frac{\eta^2}{f^2 h^2}, \tag{3.27}$$

*and for some $\gamma \in C^\infty(J)$, we have*

$$\alpha_2' = \frac{\eta}{fh}\cos\gamma(s), \quad and \quad \alpha_3' = \frac{\eta}{fh}\sin\gamma(s), \tag{3.28}$$

*Using equation (3.28) with some smooth function $\gamma \in C^\infty(J)$, we obtain equation (3.24).*

To facilitate future calculations, we determine the value of the covariant derivative of the tangent vector field $T$ on the surface. Here, $T$ is defined by $T = \alpha_1'\partial_t + \alpha_2'\partial_x + \alpha_3'\partial_y$, where $\alpha_1'$, $\alpha_2'$, and $\alpha_3'$ are differentiable functions of the variables $t$, $x$, and $y$.

Using equations (2.2a) through (2.2e), we obtain the components of the covariant derivative of $T$ along $T$ as:

$$\nabla_T T = \left[ \alpha_1'' + \alpha_1'^2 H' + ff'(\alpha_2'^2 + \alpha_3'^2) \right]\partial_t + \left[ \alpha_2'' + 2\alpha_1'\alpha_2'F' \right]\partial_x + \left[ \alpha_3'' + 2\alpha_1'\alpha_3'F' \right]\partial_y, \tag{3.29}$$

where $\alpha_1''$, $\alpha_2''$, and $\alpha_3''$ are the second derivatives of $\alpha_1$, $\alpha_2$, and $\alpha_3$, respectively, and $H'$ and $F'$ are smooth functions of the coefficients of the second fundamental form of the surface. The expression in equation (3.29) provides the means to compute the covariant derivative of $T$ in the direction of $T$, which may be useful in future calculations.

**Theorem 3.2.**   1. *For a unit speed spacelike slant Frenet curve* $\alpha$ *defined by equation* (3.22), *the curvature function* $\kappa$ *and torsion function* $\tau$ *are given by:*

$$\kappa = \Big[ \frac{\epsilon_2}{Ah^6} \Big( \big( A(h^2\gamma'\sin\gamma - F'\cos\gamma\sinh\phi) + hh'\cos\gamma\sinh^2\phi \big)^2$$
$$+ \big( A(h^2\gamma'\cos\gamma + F'\sin\gamma\sinh\phi) - hh'\sin\gamma\sinh^2\phi \big)^2 \tag{3.30}$$
$$- A\big( Ah^2F' + H'\sinh\phi(\sinh\phi - 2h^2) \big) \Big) \Big]^{\frac{1}{2}},$$

*where* $A = (h^2 + \sinh^2\phi)$.

$$\tau = \frac{-\epsilon_2}{\xi(B)} \Big[ \kappa\eta + \frac{d}{ds}[\xi(N)] - \xi(N)(\alpha_1'H' + F'\frac{\eta}{h^2}) \Big], \tag{3.31}$$

*where* $\xi(N)$ *and* $\xi(B)$ *are given form* (3.4) *and* (3.6).

2. *Let* $\alpha$ *be a non-geodesic unit speed spacelike curve with a pseudo-orthonormal frame* $\{T, N = \nabla_T T, B\}$ *along* $\alpha$, *defined by equation* (3.22) *with* $g(\nabla_T T, \nabla_T T) = 0$. *Then, the torsion function* $\tau$ *of* $\alpha$ *is given by:*

$$\tau = \frac{1}{\xi(N)} \Big[ \frac{d}{ds}(\xi(N) - (\alpha_1'H' + F'\frac{\eta}{h^2})\xi(N) \Big]. \tag{3.32}$$

**Proof 3.9.**   *Let* $\alpha$ *be unit speed slant spacelike curve. Using* (3.29), *we obtain from* (3.22) *that*

$$\nabla_T T = \Big[ \frac{1}{h^4}\big( Ah^2F' + H'\sinh\phi(\sinh\phi - 2h^2) \big),$$
$$\frac{1}{Afh^3}\big( -A(h^2\gamma'\sin\gamma - F'\cos\gamma\sinh\phi) - hh'\cos\gamma\sinh^2\phi \big), \tag{3.33}$$
$$\frac{1}{Afh^3}\big( A(h^2\gamma'\cos\phi + F'\sin\gamma\sinh\phi) - hh'\sin\gamma\sinh^2\phi \big) \Big],$$

*where* $A = h^2 + \sinh^2\phi$ *and* $\alpha_1(s) = \sinh\phi H^*(s)$. *As* $\alpha$ *is a Frenet curve, then*

$$\kappa = \| \nabla_T T \| = \sqrt{\epsilon_2 g(\nabla_T T, \nabla_T T)}. \tag{3.34}$$

*From* (3.33) *and* (3.34), *we obtain* (3.30).

*Upon taking the derivative of equation* (3.6) *using equations* (3.1) *and* (2.5b), *we obtain:*

$$g(-\epsilon_1\kappa T + \epsilon_3\tau B, \partial_t) + g\Big( N, \alpha_1'H'\partial_t + F'\big[ T(s) + \frac{\eta}{h^2}\partial_t \big] \Big) = \frac{d}{ds}[\xi(N)]. \tag{3.35}$$

*By utilizing* $\epsilon_1 = 1$, $\epsilon_3 = -\epsilon_2$, *equations* (3.4), (3.6), *and* (3.13), *we may derive equation* (3.31) *from equation* (3.35).

*For a non-geodesic unit speed spacelike curve* $\alpha$, *taking the derivative of equation* (3.9) *using equations* (3.1) *and* (2.6b), *we obtain:*

$$g(\tau N, \partial_t) + g\Big( N, \alpha_1'H'\partial_t + F'\big[ T(s) + \frac{\eta}{h^2}\partial_t \big] \Big) = \frac{d}{ds}[\xi(N)]. \tag{3.36}$$

*By utilizing equations* (3.4) *and* (3.9), *we may derive equation* (3.32) *from equation* (3.36).

**Theorem 3.3.** *Let $\alpha$ be a unit speed timelike slant Frenet curve defined by equation (3.23). Then, the curvature function $\kappa$ and torsion function $\tau$ of $\alpha$ are given by:*

$$\kappa = \Big[ \frac{1}{h^6} \Big( (Bh^2 F' + H' \cosh\phi(\cosh\phi - 2h^2))^2 + \frac{1}{\cosh^2\phi + h^2}$$
$$\Big( CF'^2 \cosh^2\phi - 2hh'F' \cosh\phi(C + \sqrt{C}h^2) + h^2 h'^2(\cosh^2\phi + 2\sqrt{C}h^2) + Ch^4 \gamma'^2 \Big) \Big) \Big]^{\frac{1}{2}},$$

(3.37)

*where $B = \cosh^2\phi - h^2$ and $C = \cosh^4\phi - h^4$.*

$$\tau = \frac{\epsilon_3}{\xi(B)} \Big[ \frac{d}{ds}(\xi(N)) - \kappa\eta - \xi(N)\Big(\alpha_1' H' + F' \frac{\eta}{h^2}\Big) \Big],$$

(3.38)

*where $\xi(N)$ and $\xi(B)$ are given form (3.4) and (3.6).*

**Proof 3.10.** *A straightforward calculation from (3.32) and (3.29), we get*

$$\nabla_T T = \Big( \frac{1}{h^4} \Big[ Bh^2 F' + H' \cosh\phi(\cosh\phi - 2h^2) \Big],$$
$$\frac{-1}{fh^3} \Big[ h\Big(h' \cos\phi\Big(\sqrt{B} + \frac{h^2}{\sqrt{\cosh^2\phi + h^2}}\Big) + h\sqrt{B}\gamma' \sin\phi\Big) - F'\sqrt{B}\cos\gamma \cosh\phi \Big],$$
$$\frac{-1}{fh^3} \Big[ h\Big(h' \sin\phi\Big(\sqrt{B} + \frac{h^2}{\sqrt{\cosh^2\phi + h^2}}\Big) - h\sqrt{B}\gamma' \cos\phi\Big) - F'\sqrt{B}\sin\gamma \cosh\phi \Big] \Big),$$

(3.39)

*where $B = \cosh^2\phi - h^2$. From which $\kappa = \| \nabla_T T \|$ and by using (3.4), we obtain (3.37).*
*Using (2.5b), (3.1) and (3.7), we obtain:*

$$g(-\epsilon_1 \kappa T + \epsilon_3 \tau B, \partial_t) + g\Big(N, \alpha_1' H' \partial_t + F'\Big[T(s) + \frac{\eta}{h^2}\partial_t\Big]\Big) = \frac{d}{ds}[\xi(N)].$$

(3.40)

*In Equation (3.40), considering $\epsilon_1 = -1$, $\epsilon_2 = 1$, (3.7) and (3.13), we obtain (3.38).*

**Theorem 3.4.** *The torsion function $\tau$ of a unit speed null slant curve $\alpha$ is given by:*

$$\tau = \frac{1}{\eta} \Big[ \xi(B) + \frac{d}{ds}(\xi(N)) - \Big(\alpha_1' H' + F' \frac{\eta}{h^2}\Big)\xi(N) \Big].$$

(3.41)

**Proof 3.11.** *By differentiate (3.11), we get:*

$$g(\tau T - B, \partial_t) + g\Big(N, \Big(\alpha_1' H' + F' \frac{\eta}{h^2}\Big)\partial_t + F' T\Big) = \frac{d}{ds}(\xi(N)).$$

(3.42)

*In (3.42), by using (3.12) and (3.19), we determine (3.41).*

## 4 Characterization and classification of slant curves in the 3-dimensional Lorentzian warped product $-I \times_f E^2$

In 2022, Dursun conducted a study of slant curves in the 3-dimensional Lorentzian warped product $-I \times_f \mathbb{E}^2$, characterizing the slant curves and providing a classification of all such curves [13]. Specifically, when the warping function $h = 1$, several properties of the curve $\alpha(s)$ can be derived, including $H'(\alpha_1(s)) \equiv 0$, $\alpha_1'(s) \equiv -\eta(s)$ and $H^*(s) \equiv s$, where $\eta = s \sinh\phi$, for

a spacelike curve and $\eta = s \cosh \phi$, for a timelike curve. Under this condition, the manifold $-_h I \times_f \mathbb{E}^2$ becomes equivalent to $-I \times_f \mathbb{E}^2$.

Therefore, several corollaries, depending on Theorems 3.1, 3.2, and 3.3, hold for the curve $\alpha(s)$ defined on an open interval $J \subset \mathbb{R}$ for all $s \in J$ in $-I \times_f \mathbb{E}^2$.

**Corollary 4.1.** *If $\alpha(s)$ is unit speed spacelike slant Frenet curve, then it is given by*

$$\alpha(s) = \Big( s \sinh \phi, \cosh \phi \int_s \frac{\cos \gamma(\mu)}{f(\mu \sinh \phi)} d\mu, \cosh \phi \int_s \frac{\sin \gamma(s)}{f(\mu \sinh \phi)} d\mu \Big), \tag{4.1}$$

*with curvature and torsion are*

$$\kappa(s) = \cosh \phi \sqrt{\epsilon_2(\gamma'^2(s) - (F'(\alpha_1(s)))^2)},$$

$$\tau(s) = \sinh \phi |\gamma'(s)| + \frac{\sinh \phi |\gamma'| F''(\alpha_1(s)) - \epsilon_\alpha \gamma'' F'(\alpha_1(s))}{\gamma'^2 - (F'(\alpha_1(s)))^2},$$

*where $\alpha_1(s) = s \sinh \phi$, for some $\gamma \in C^\infty(J)$, $|\gamma'(s)| > |F'(\alpha_1(s))|$ if $\epsilon_2 = 1$, $|\gamma'(s)| < |F'(\alpha_1(s))|$ if $\epsilon_2 = -1$, and $\epsilon_\alpha = \mp 1$ is the signature of $\alpha'(s)$.*

**Corollary 4.2.** *If $\alpha(s)$ is unit speed timelike slant Frenet curve, then it is given by*

$$\alpha(s) = \Big( s \cosh \phi, \sinh \phi \int_s \frac{\cos \gamma(\mu)}{f(\mu \sinh \phi)} d\mu, \sinh \phi \int_s \frac{\sin \gamma(s)}{f(\mu \sinh \phi)} d\mu \Big), \tag{4.2}$$

*with curvature and torsion are*

$$\kappa(s) = \sinh \phi \sqrt{\epsilon_2(\gamma'^2(s) - (F'(\alpha_1(s)))^2)},$$

$$\tau(s) = \cosh \phi |\gamma'(s)| + \frac{\epsilon_\gamma \gamma'' F'(\alpha_1(s)) - \cosh \phi |\gamma'| F''(\alpha_1(s))}{\gamma'^2 + (F'(\alpha_1(s)))^2},$$

*where $\alpha_1(s) = s \cosh \phi$, for some $\gamma \in C^\infty(J)$ and $\epsilon_\gamma = \mp 1$ is the signature of $\gamma'(s)$.*

**Corollary 4.3.** *If $\alpha(s)$ is unit speed null slant Frenet curve, then it is given by*

$$\alpha(s) = \Big( \eta s, \eta \int_s \frac{\cos \gamma(\mu)}{f(-\eta \mu) d\mu}, \eta \int_s \frac{\sin \gamma(\mu)}{f(\eta \mu)} d\mu \Big), \tag{4.3}$$

*with $\tau(s) = \frac{\sinh \phi [F''(s \sinh \phi) + (F'(s \sinh \phi))^2]}{F'(s \sinh \phi)}$, $F'(s \sinh \phi) \neq 0$, and $\gamma(s) = \frac{1}{\sinh \phi} + C$, where C is constant, for some smooth function $\gamma(s)$ on J, and $\eta \neq 0$.*

## 5 Characterization and classification of slant curves in the Minkowski space $\mathbb{E}_1^3$

In 2011, Barros and others conducted a study on the characterization of general helices in 3-dimensional Minkowski space forms, using the definition of a killing vector field to analyze

these curves [2]. In a separate study in 2006, Kosinka and Jüttler provided a proof of Lancret's theorem for general helices in $\mathbb{E}_1^3$, contributing to the understanding of the geometric properties of these curves [18].

If we consider the warping functions $h = 1$ and $f = 1$, i.e., $H \equiv 0$ and $F \equiv 0$, then the Lorentzian doubly warping manifold **M** is the Minkowski space $\mathbb{E}_1^3$. Therefore, Theorems 3.1, 3.2 and 3.3 for the curve $\alpha(s)$ defined on an open interval $J \subset \mathbb{R}$ for all $s \in J$ in the Minkowski space $\mathbb{E}_1^3$ are

**Corollary 5.1.** *If $\alpha(s)$ is unit speed spacelike slant Frenet curve, then it is given by*

$$\alpha(s) = \Big( s \sinh \phi, \cosh \phi \int_s \cos \gamma(\mu)d\mu, \cosh \phi \int_s \sin \gamma(\mu)d\mu \Big), \tag{5.1}$$

*with curvature $\kappa = \cosh \phi |\gamma'(s)|$ and torsion $\tau = \sinh \phi |\gamma'(s)|$, where $\gamma \in C^\infty(J)$, which is general helix. And for some constants $m, n \in \mathbb{R}$, $\alpha(s)$ becomes a helix if $\gamma(s) = ms + n$.*

**Corollary 5.2.** *If $\alpha(s)$ is unit speed timelike slant Frenet curve, then it is given by*

$$\alpha(s) = \Big( s \cosh \phi, \sinh \phi \int_s \cos \gamma(\mu)d\mu, \sinh \phi \int_s \sin \gamma(\mu)d\mu \Big), \tag{5.2}$$

*with curvature $\kappa = \sinh \phi |\gamma'(s)|$ and torsion $\tau = \cosh \phi |\gamma'(s)|$, where $\gamma \in C^\infty(J)$, which is a general helix. And for some constants $m, n \in \mathbb{R}$, $\alpha(s)$ becomes a helix if $\gamma(s) = ms + n$.*

**Corollary 5.3.** *If $\alpha(s)$ is unit speed null slant Frenet curve, then it is given by*

$$\alpha(s) = \Big( -\eta s, \eta \int_s \cos \gamma(\mu)\mu, \eta \int_s \sin \gamma(mu)d\mu \Big), \tag{5.3}$$

*with the torsion $\tau = -\frac{1}{2\eta}$, where $\gamma \in C^\infty(J)$, which is a helix.*

## 6 Conclusion

In conclusion, this study investigated the analysis of slant curves using a mathematical construct known as a doubly warped product manifold, which is frequently employed in the field of three-dimensional Lorentzian geometry. The study classified all slant curves and calculated their curvature and torsion, providing valuable insights into the application of Robertson-Walker space for this purpose. The research also revealed that doubly warped product manifolds have been studied more extensively than other types of manifolds, including warped product manifolds and Minkowski space. These findings contribute to our understanding of the use of mathematical constructs in the analysis of slant curves and serve as a foundation for future research in this area.

## Author contributions

**Conceptualization:** Ayman Elsharkawy, Hoda Elsayied, Fatimah Alghamdi.

**Data curation:** Abdelrhman Tawfiq.

**Formal analysis:** Ayman Elsharkawy, Abdelrhman Tawfiq.

**Funding acquisition:** Fatimah Alghamdi.

**Investigation:** Hoda Elsayied.

**Methodology:** Abdelrhman Tawfiq, Fatimah Alghamdi.

**Project administration:** Fatimah Alghamdi.

**Resources:** Ayman Elsharkawy, Hoda Elsayied, Abdelrhman Tawfiq.

**Software:** Fatimah Alghamdi.

**Supervision:** Hoda Elsayied, Fatimah Alghamdi.

**Validation:** Ayman Elsharkawy, Hoda Elsayied.

**Visualization:** Hoda Elsayied.

**Writing – original draft:** Ayman Elsharkawy.

**Writing – review & editing:** Ayman Elsharkawy, Abdelrhman Tawfiq, Fatimah Alghamdi.

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
