## [Decision Letter · Decision Letter 0]

13 Feb 2025

PONE-D-24-52708Investigating Slant Curves within Lorentzian Doubly Warped Product ManifoldsPLOS ONE

Dear Dr. Elsharkawy,

Thank you for submitting your manuscript to PLOS ONE. After careful consideration, we feel that it has merit but does not fully meet PLOS ONE’s publication criteria as it currently stands. Therefore, we invite you to submit a revised version of the manuscript that addresses the points raised during the review process.

We look forward to receiving your revised manuscript.

Kind regards,

Pengpeng Hu

Academic Editor

PLOS ONE

Reviewers' comments:

Reviewer's Responses to Questions

**Comments to the Author**

1. Is the manuscript technically sound, and do the data support the conclusions?

Reviewer #1: Yes

Reviewer #2: Yes

2. Has the statistical analysis been performed appropriately and rigorously? 

Reviewer #1: N/A

Reviewer #2: N/A

3. Have the authors made all data underlying the findings in their manuscript fully available?

Reviewer #1: Yes

Reviewer #2: Yes

4. Is the manuscript presented in an intelligible fashion and written in standard English?

Reviewer #1: Yes

Reviewer #2: Yes

5. Review Comments to the Author

Reviewer #1: The manuscript entitled Investigating Slant Curves within Lorentzian Doubly Warped Product Manifolds” proposes new theorems and concept regarding the slant curves by applying of Robertson-Walker space.

This paper is a pure mathematics in differential geometry field. Overall, this paper for has no major errors in terms of proving the theorems and lemmas. In terms of theoretical, this paper has significant and interesting result by providing analysis of the slant curves within 3D Lorentzian doubly warped product manifold.

There is some minor mistakes in this paper such as:

1. Page 2, paragraph 4, please state the full name of acronym GRW.

2. Page 6, Proposition 3.1, please capitalized the beginning word "if" in point 1 and 2.

Hence, I have no objection for accepting this paper if the author amend the minor mistakes.

Reviewer #2: The work is thorough and largely very well-written -- I wish to commend the authors on their clear writing and accessible presentation of the material. The scope of the work is limited, and the authors tell its story rigorously and effectively without sacrificing clarity. While I do have some suggestions to improve the paper, I think this work is strong overall and suitable for publication after revision.

(please see attachment for detailed review notes)

6. PLOS authors have the option to publish the peer review history of their article (what does this mean?). If published, this will include your full peer review and any attached files.

Reviewer #1: No

Reviewer #2: **Yes: **Ty O Easley

---

## [Author Response · Author response to Decision Letter 1]

18 Feb 2025

Dear Editor,

I would like to express my sincere gratitude for the time and effort you have dedicated to revising our manuscript. We truly appreciate your thorough examination of every detail in the study to ensure its accuracy and readiness for publication in Plos one. Your careful attention to each aspect of the manuscript has been invaluable in improving the overall quality of our work.

In addition, we would like to thank you for your insightful suggestions, which have helped us refine our analysis and strengthen the clarity of the paper. We are confident that the revisions made in response to your feedback, as well as the feedback from the reviewers, have significantly enhanced the manuscript.

As requested, we have carefully addressed each referee’s comment in detail. Please find attached the response letter, which outlines how we have incorporated their suggestions and made the necessary revisions.

R eviewer 1

We have carefully considered each of your comments and have provided a detailed response to ensure that all necessary revisions have been implemented effectively:

Comment 1.

Page 2, paragraph 4, please state the full name of acronym GRW.

Response: We defined the acronym as Generalized Robertson–Walker (GRW).

Comment 2.

Page 6, Proposition 3.1, please capitalized the beginning word "if" in point 1 and 2.

Response: We made the required.

R eviewer 2

We sincerely appreciate your dedicated time and effort in reviewing our manuscript. Your insightful comments and precise observations have greatly improved our work's quality and clarity. It is evident that you are deeply engaged with the latest developments in this field, and we are grateful for your thorough and thoughtful assessment.

Your feedback has helped us refine key aspects of our study, and we have carefully addressed each of your suggestions in the detailed responses below. We believe that these revisions have strengthened the manuscript significantly, and we truly value your expertise in guiding us toward a clearer and more robust presentation of our findings.

Please find below our point-by-point responses to your comments.

Part I. Frenet-Serret equations

Comment 1.

why the Frenet-Serret equations should take the form of equations (2.6*) and (2.7*) in the spacelike and null curve cases,

Response

We added the following reference:

Walrave, Johan. Curves and Surfaces in Minkowski Space. PhD diss., Doctoral thesis, KU Leuven, Faculty of Science, Leuven, 1995.

This work discusses in detail how to derive the Frenet-Serret equations in all cases.

Comment 2.

Why would this be a lightlike curve and not a spacelike one? If it is a null curve, how does a spacelike covariant derivative relate to the vanishing norm of the tangent vector?

Response

If the tangent vector α' were spacelike, then g(α',α')>0, which contradicts the assumption that the curve is null. The spacelike nature of ∇_α' α' does not affect the fact that α' is always lightlike; it merely indicates that the curve is not a geodesic and that there is some nontrivial acceleration in a spacelike direction.

A curve is classified as null based on its tangent vector α', not its acceleration. The fact that ∇_α' α' is spacelike means the curve is accelerating in a spacelike direction, but the curve itself remains lightlike because its tangent vector is always null.

Comment 3.

Why does the paragraph beginning ”A non-null curve α(s) [...]” not instead say ”A timelike curve α(s) [...]”? This definition is only used in the timelike case later. (Also, I think it may be clearer to swap this paragraph with the previous one.)

Response

When discussing non-null curves in Minkowski space, we classify them into three types based on the causal nature of the tangent, normal, and binormal vectors:

(Spacelike, Spacelike, Timelike)

(Timelike, Spacelike, Spacelike)

(Spacelike, Timelike, Spacelike)

These classifications provide a generalized framework for studying non-null curves. Depending on the specific study, we select the appropriate type based on the properties required for analysis.

Comment 4.

Why is it a desired property that the norm of the covariant derivative of a Frenet curve be identically 0 if that curve is spacelike and nonvanishing if that curve is timelike?

Response

The desired property, based on the norm of ∇_α' α', can be summarized in the following cases:

Curve Type Norm of ∇_α' α' Interpretation

Spacelike Frenet Curve Zero (Spacelike) Represents a geodesic (straightest path in a spacelike direction).

Spacelike Frenet Curve Non-Zero (Timelike or Spacelike) Represents a non-geodesic motion with curvature.

Timelike Frenet Curve Zero (Timelike Represents a timelike geodesic (free-fall motion).

Timelike Frenet Curve Non-Zero (Spacelike) Represents an accelerating massive particle (non-geodesic motion).

Note: The appropriate case is chosen based on the specific focus of our study in this part of the manuscript.

Comment 5.

In equations (2.6a-d), is it still the case that T = α'(s)? If not, what is the relationship between α(s) and the rest of the Frenet apparatus?

Response

It still holds that T = α^' (s) as the tangent vector.

Comment 6.

Lastly, this is a smaller issue, but I would also recommend writing out the Frenet curve definition as its own definition (e.g., Defintion 2.3), or at least typographically offsetting it in some way that makes it easier to refer back to.

Response

We added the definition of the Frenet curve at the beginning of Section 2.3, Frenet-Serret Equations.

Part II. Style

Comment 1.

This is mostly a cosmetic complaint, but it would also help with clarity: please state common theorem data before enumerating cases in Theorems 3.1 and 3.2. (e.g., ”Let α(s) be a unit-speed slant Frenet curve. (a) if α(s) timelike, then… (b) if α(s) spacelike, then ... (c) if α(s) null, then ... etc)

Response

We stated a common theorem, making the presentation more scientific and clearer.

Comment 2.

Additionally, please state that γ∈C^∞ (J) somewhere in the theorem statement.

Response

We added it at the end of the theorem's statement.

Comment 3.

Finally, Corollaries 4.1 and 5.1 would be much more readable if separated into individual corollaries.

Response

We separated them into three separate corollaries.

Part III. Line items

Comment 1.

In the last paragraph of the introduction, the acronym ”GRW” is used without definition, and I don’t know what it means.

Response

We defined the acronym as Generalized Robertson–Walker (GRW).

Comment 2.

Equation (2.1a) should read g(∇_(∂_t ) ∂_t,∂_t ), and it might be worth explicitly mentioning that this vanishes specifically because h does not depend on t (per Lemma 3.1 of Du and Wang’s cited paper)

Response

There was a typo, and we have fixed it to be like g(∇_(∂_t ) ∂_t,∂_t ).

Comment 3.

The ϵ_i equations (2.5a-e) are labeled as ”casual characters” and I think should be ”causal characters.”

Response

There was a typo, and we have fixed it to be “causal characters”.

Comment 4.

In the first paragraph of section 3, please provide citations illustrating that ”slant curves have signifcant implications in fields such as computer graphics, robotics, and

biophysics.”

Response

We have added a reference that highlights the significant implications of slant curves in the study of these fields.

Comment 5.

In the line following equation 3.5, ”Form” should be ”From” and I’m confused as to why h(α_1 ) would hold instead of h(α_2 (s),α_3 (s))

Response

There was a typo, and we have fixed it as h is defined as h(α_2 (s),α_3 (s)).

Comment 6.

In the first line of the proof of Proposition 3.5, g(T,∂_t,T) is written (which of course is not an expression that makes sense), so it’s unclear why the first term of equation (3.17) is ηT.

Response

There was a typo, and we have fixed it as g(T,∂_t).

The first term of Equation (3.17) is ηT. It follows directly from Equation (3.5) after differentiating the curve α(s), taking the dot product of α^' (s)with ∂_t according to the defined metric, and setting H^*' = 1 for simplicity. This yields the desired outcome.

Comment 7.

A few times in sections 3 and 4, the phrase ”for some α∈C^∞ (J) shows up – it seems like α should be γ in these lines, but it would be easier to be certain if data were clearly defined in the statement of theorems and propositions.

Response

There were typos, and we have fixed them. We also revised them in these two sections.

Once again, we sincerely thank you for your time and effort in reviewing our manuscript. Your precise and insightful comments have significantly improved the quality and clarity of our work. We truly appreciate your valuable contributions to this revision.

Sincerely,

The authors

---

## [Editor Report · Decision Letter 1]

24 Feb 2025

Investigating Slant Curves within Lorentzian Doubly Warped

Product Manifolds

PONE-D-24-52708R1

Dear Dr. Elsharkawy,

We’re pleased to inform you that your manuscript has been judged scientifically suitable for publication and will be formally accepted for publication once it meets all outstanding technical requirements.

Kind regards,

Pengpeng Hu

Academic Editor

PLOS ONE

---

## [Editor Report · Acceptance letter]

PONE-D-24-52708R1

PLOS ONE

Dear Dr. Elsharkawy,

I'm pleased to inform you that your manuscript has been deemed suitable for publication in PLOS ONE. Congratulations! Your manuscript is now being handed over to our production team.

Kind regards,

on behalf of

Dr. Pengpeng Hu

Academic Editor

PLOS ONE